# ZERO-SHOT VISUAL IMITATION

**Deepak Pathak**[*], **Parsa Mahmoudieh**[*], **Guanghao Luo**[*], **Pulkit Agrawal**[*], **Dian Chen**,
**Yide Shentu**, **Evan Shelhamer**, **Jitendra Malik**, **Alexei A. Efros**, **Trevor Darrell**

UC Berkeley
{pathak,parsa.m,michaelluo,pulkitag,dianchen,
fredshentu,shelhamer,malik,efros,trevor}@cs.berkeley.edu

## ABSTRACT

The current dominant paradigm for imitation learning relies on strong supervision of expert actions to learn both *what* and *how* to imitate. We pursue an alternative paradigm wherein an agent first explores the world without any expert supervision and then distills its experience into a goal-conditioned skill policy with a novel forward consistency loss. In our framework, the role of the expert is only to communicate the goals (i.e., what to imitate) during inference. The learned policy is then employed to mimic the expert (i.e., how to imitate) after seeing just a sequence of images demonstrating the desired task. Our method is "zero-shot" in the sense that the agent never has access to expert actions during training or for the task demonstration at inference. We evaluate our zero-shot imitator in two real-world settings: complex rope manipulation with a Baxter robot and navigation in previously unseen office environments with a TurtleBot. Through further experiments in VizDoom simulation, we provide evidence that better mechanisms for exploration lead to learning a more capable policy which in turn improves end task performance. Videos, models, and more details are available at https://pathak22.github.io/zeroshot-imitation/.

## 1 INTRODUCTION

Imitating expert demonstration is a powerful mechanism for learning to perform tasks from raw sensory observations. The current dominant paradigm in learning from demonstration (LfD) (Argall et al., 2009; Ng & Russell, 2000; Pomerleau, 1989; Schaal, 1999) requires the expert to either manually move the robot joints (i.e., kinesthetic teaching) or teleoperate the robot to execute the desired task. The expert typically provides multiple demonstrations of a task at training time, and this generates data in the form of observation-action pairs from the agent's point of view. The agent then distills this data into a policy for performing the task of interest. Such a heavily supervised approach, where it is necessary to provide demonstrations by controlling the robot, is incredibly tedious for the human expert. Moreover, for every new task that the robot needs to execute, the expert is required to provide a new set of demonstrations.

Instead of communicating *how* to perform a task via observation-action pairs, a more general formulation allows the expert to communicate only *what* needs to be done by providing the observations of the desired world states via a video or a sparse sequence of images. This way, the agent is required to infer how to perform the task (i.e., actions) by itself. In psychology, this is known as *observational learning* (Bandura & Walters, 1977). While this is a harder learning problem, it is a more interesting setting, because the expert can demonstrate multiple tasks quickly and easily.

An agent without any prior knowledge will find it extremely hard to imitate a task by simply watching a visual demonstration in all but the simplest of cases. Thus, the natural question is: in order to imitate, what form of prior knowledge must the agent possess? A large body of work (Breazeal & Scassellati, 2002; Dillmann, 2004; Ikeuchi & Suehiro, 1994; Kuniyoshi et al., 1989; 1994; Yang et al., 2015) has sought to capture prior knowledge by manually pre-defining the state that must be

---

[*]Denotes equal contribution.

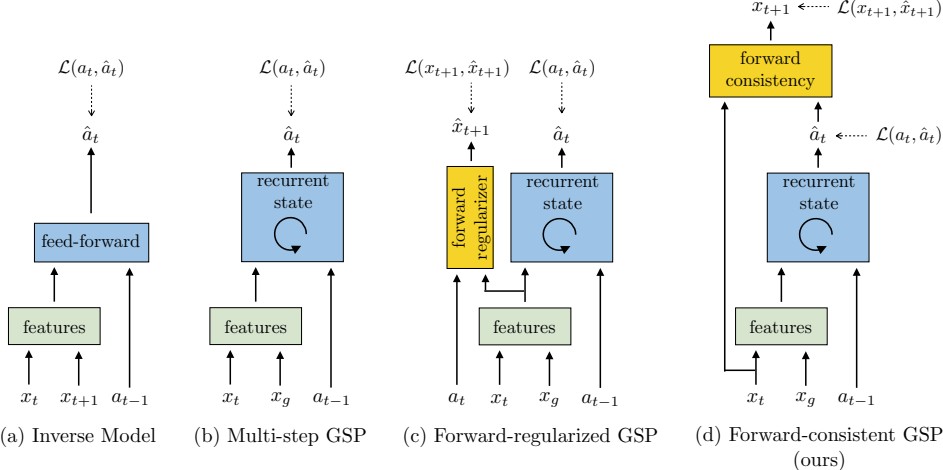

Figure 1: The goal-conditioned skill policy (GSP) takes as input the current and goal observations and outputs an action sequence that would lead to that goal. We compare the performance of the following GSP models: (a) Simple inverse model; (b) Mutli-step GSP with previous action history; (c) Mutli-step GSP with previous action history and a forward model as regularizer, but no forward consistency; (d) Mutli-step GSP with forward consistency loss proposed in this work.

inferred from the observations. The agent then infers how to perform the task (i.e., plan for imitation) using this state. Unfortunately, computer vision systems are often unable to estimate the state variables accurately and it has proven non-trivial for downstream planning systems to be robust to such errors.

In this paper, we follow (Agrawal et al., 2016; Levine et al., 2016; Pinto & Gupta, 2016) in pursuing an alternative paradigm, where an agent explores the environment without any expert supervision and distills this exploration data into goal-directed skills. These skills can then be used to imitate the visual demonstration provided by the expert (Nair et al., 2017). Here, by skill we mean a function that predicts the sequence of actions to take the agent from the current observation to the goal. We call this function a *goal-conditioned skill policy (GSP)*. The GSP is learned in a self-supervised way by re-labeling the states visited during the agent's exploration of the environment as goals and the actions executed by the agent as the prediction targets, similar to (Agrawal et al., 2016; Andrychowicz et al., 2017). During inference, given goal observations from a demonstration, the GSP can infer how to reach these goals in turn from the current observation, and thereby imitate the task step-by-step.

One critical challenge in learning the GSP is that, in general, there are multiple possible ways of going from one state to another: that is, the distribution of trajectories between states is multi-modal. We address this issue with our novel *forward consistency loss* based on the intuition that, for most tasks, reaching the goal is more important than how it is reached. To operationalize this, we first learn a forward model that predicts the next observation given an action and a current observation. We use the difference in the output of the forward model for the GSP-selected action and the ground truth next state to train the GSP. This loss has the effect of making the GSP-predicted action *consistent* with the ground-truth action instead of exactly matching the actions themselves, thus ensuring that actions that are different from the ground-truth—but lead to the same next state— are not inadvertently penalized. To account for varying number of steps required to reach different goals, we propose to jointly optimize the GSP with a goal recognizer that determines if the current goal has been satisfied. See Figure 1 for a schematic illustration of the GSP architecture.

We call our method *zero-shot* because the agent never has access to expert actions, neither during training of the GSP nor for task demonstration at inference. In contrast, most recent work on one-shot imitation learning requires full knowledge of actions and a wealth of expert demonstrations during training (Duan et al., 2017; Finn et al., 2017). In summary, we propose a method that (1) does not require any extrinsic reward or expert supervision during learning, (2) only needs demonstrations during inference, and (3) restricts demonstrations to visual observations alone rather than full state-actions. Instead of learning by imitation, our agent learns to imitate.

We evaluate our zero-shot imitator on real-world robots for rope manipulation tasks using a Baxter and office navigation using a TurtleBot. We show that the proposed forward consistency loss improves the performance on the complex task of knot tying from $36\%$ to $60\%$ accuracy. In navigation experiments, we steer a simple wheeled robot around partially-observable office environments and show that the learned GSP generalizes to unseen environments. Furthermore, using navigation experiments in *VizDoom* environment, we show that (GSP) learned using curiosity-driven exploration (Oudeyer et al., 2007; Pathak et al., 2017; Schmidhuber, 1991) can more accurately follow demonstrations as compared to using random exploration data for learning the GSP. Overall our experiments show that the forward-consistent GSP can be used to imitate a variety of tasks without making environment or task-specific assumptions.

## 2 LEARNING TO IMITATE WITHOUT EXPERT SUPERVISION

Let $\mathcal{S} : \{x_1, a_1, x_2, a_2, ..., x_T\}$ be the sequence of observations and actions generated by the agent as it explores its environment using the policy $a = \pi_E(s)$. This exploration data is used to learn the goal-conditioned skill policy (GSP) $\pi$ takes as input a pair of observations $(x_i, x_g)$ and outputs sequence of actions $(\vec{a}_\tau : a_1, a_2 ... a_K)$ required to reach the goal observation $(x_g)$ from the current observation $(x_i)$.

$$\vec{a}_\tau = \pi(x_i, x_g; \theta_\pi) \tag{1}$$

where states $x_i, x_g$ are sampled from the $\mathcal{S}$. The number of actions, $K$, is also inferred by the model. We represent $\pi$ by a deep network with parameters $\theta_\pi$ in order to capture complex mappings from visual observations $(x)$ to actions. $\pi$ can be thought of as a variable-step generalization of the inverse dynamics model (Jordan & Rumelhart, 1992), or as the policy corresponding to a universal value function (Foster & Dayan, 2002; Schaul et al., 2015), with the difference that $x_g$ need not be the end goal of a task but can also be an intermediate sub-goal.

Let the task to be imitated be provided as a sequence of images $\mathcal{D} : \{x_1^d, x_2^d, ..., x_N^d\}$ captured when the expert demonstrates the task. This sequence of images $\mathcal{D}$ could either be temporally dense or sparse. Our agent uses the learned GSP $\pi$ to imitate the sequence of visual observations $\mathcal{D}$ starting from its initial state $x_0$ by following actions predicted by $\pi(x_0, x_1^d; \theta_\pi)$. Let the observation after executing the predicted action be $x_0'$. Since multiple actions might be required to reach close to $x_1^d$, the agent queries a separate *goal recognizer* network to ascertain if the current observation is close to the goal or not. If the answer is negative, the agent executes the action $a = \pi(x_0', x_1^d; \theta_\pi)$. This process is repeated iteratively until the *goal recognizer* outputs that agent is near the goal, or a maximum number of steps are reached. Let the observation of the agent at this point be $\hat{x}_1$. After reaching close to the first observation $(x_1^d)$ in the demonstration, the agent sets its goal as $(x_2^d)$ and repeats the process. The agent stops when all observations in the demonstrations are processed.

Note that in the method of imitation described above, the expert is never required to convey to the agent what actions it performed. In the following subsections we describe how we learn the GSP, *forward consistency loss*, *goal recognizer* network and various baseline methods.

### 2.1 LEARNING THE GOAL-CONDITIONED SKILL POLICY (GSP)

We first describe the one-step version of GSP and then extend it to variable length multi-step skills. One-step trajectories take the form of $(x_t, a_t, x_{t+1})$ and GSP, $\hat{a}_t = \pi(x_t, x_{t+1}; \theta_\pi)$, is trained by minimizing the standard cross-entropy loss $\mathcal{L}(a_t, \hat{a}_t)$,

$$\mathcal{L}(a_t, \hat{a}_t) = p(a_t | x_t, x_{t+1}) \log(\hat{a}_t) \tag{2}$$

with respect to parameters $\theta_\pi$, where $p$ and $\hat{a}_t$ are the ground-truth and predicted action distributions. While we do not have access to true $p$, we empirically approximate it using samples from the distribution, $a_t$, that are executed by the agent during exploration. For minimizing the cross-entropy loss, it is common to assume $p$ as a delta function at $a_t$. However, this assumption is notably violated if $p$ is inherently multi-modal and high-dimensional. If we optimize say a deep neural network assuming $p$ to be a delta function, the same inputs will be presented with different targets (due to multi-modality) leading to high-variance in gradients which in turn would make learning challenging.

In our setup, such multi-modality can occur because multiple actions can lead the agent to the same future observation from the initial observation. For instance, in navigation, if the agent is stuck against a corner, turning or moving forward all collapse to the same effect. The issue of multi-modality becomes more critical as the length of trajectories grow, because more and more paths may take the agent from the initial observation to the goal observation given more time. Furthermore, it would require many samples to even obtain a good empirical estimate of a high-dimensional multi-modal action distribution $p$.

## 2.2 FORWARD CONSISTENCY LOSS

One way to account for multi-modality is by employing the likes of variational auto-encoders (Kingma & Welling, 2013; Rezende et al., 2014). However, in many practical situations it is not feasible to obtain ample data for each mode. In this work, we propose an alternative based on the insight that in many scenarios, we only care about whether the agent reached the final state or not and the exact trajectory is of lesser interest. Instead of penalizing the actions predicted by the GSP to match the ground truth, we propose to learn the parameters of GSP by minimizing the distance between observation $\hat{x}_{t+1}$ resulting by executing the predicted action $\hat{a}_t = \pi(x_t, x_{t+1}; \theta_\pi)$ and the observation $x_{t+1}$, which is the result of executing the ground truth action $a_t$ being used to train the GSP. In this formulation, even if the predicted and ground-truth action are different, the predicted action will not be penalized if it leads to the same next state as the ground-truth action. While this formulation will not explicitly maintain all modes of the action distribution, it will reduce the variance in gradients and thus help learning. We call this penalty the *forward consistency loss*.

Note that it is not immediately obvious as to how to operationalize *forward consistency loss* for two reasons: (a) we need the access to a good *forward dynamics* model that can reliably predict the effect of an action (i.e., the next observation state) given the current observation state, and (b) such a dynamics model should be differentiable in order to train the GSP using the state prediction error. Both of these issues could be resolved if an analytic formulation of forward dynamics is known.

In many scenarios of interest, especially if states are represented as images, an analytic forward model is not available. In this work, we learn the forward dynamics $f$ model from the data, and is defined as $\tilde{x}_{t+1} = f(x_t, a_t; \theta_f)$. Let $\hat{x}_{t+1} = f(x_t, \hat{a}_t; \theta_f)$ be the state prediction for the action predicted by $\pi$. Because the forward model is not analytic and learned from data, in general, there is no guarantee that $\tilde{x}_{t+1} = \hat{x}_{t+1}$, even though executing these two actions, $a_t, \hat{a}_t$, in the real-world will have the same effect. In order to make the outcome of action predicted by the GSP and the ground-truth action to be *consistent* with each other, we include an additional term, $\|x_{t+1} - \hat{x}_{t+1}\|_2^2$ in our loss function and infer the parameters $\theta_f$ by minimizing $\|x_{t+1} - \tilde{x}_{t+1}\|_2^2 + \lambda\|x_{t+1} - \hat{x}_{t+1}\|_2^2$, where $\lambda$ is a scalar hyper-parameter. The first term ensures that the learned forward model explains ground truth transitions $(x_t, a_t, x_{t+1})$ collected by the agent and the second term ensures consistency. The joint objective for training GSP with forward model consistency is:

$$\min_{\theta_\pi, \theta_f} \|x_{t+1} - \tilde{x}_{t+1}\|_2^2 + \lambda\|x_{t+1} - \hat{x}_{t+1}\|_2^2 + \mathcal{L}(a_t, \hat{a}_t) \tag{3}$$

$$\text{s.t.} \quad \tilde{x}_{t+1} = f(x_t, a_t; \theta_f)$$
$$\hat{x}_{t+1} = f(x_t, \hat{a}_t; \theta_f)$$
$$\hat{a}_t = \pi(x_t, x_{t+1}; \theta_\pi)$$

Note that learning $\theta_\pi, \theta_f$ jointly from scratch is precarious, because the forward model $f$ might not be good in the beginning, and hence could make the gradient updates noisier for $\pi$. To address this issue, we first pre-train the forward model with only the first term and GSP separately by blocking the gradient flow and then fine-tune jointly.

**Generalization to feature space dynamics** Past work has shown that learning forward dynamics in the feature space as opposed to raw observation space is more robust and leads to better generalization (Agrawal et al., 2016; Pathak et al., 2017). Following these works, we extend the GSP to make predictions in feature representation $\phi(x_t), \phi(x_{t+1})$ of the observations $x_t, x_{t+1}$ respectively learned through the self-supervised task of action prediction. The forward consistency loss is then computed by making predictions in this feature space $\phi$ instead of raw observations. The optimization objective for feature space generalization with mutli-step objective is shown in Equation (4).

**Generalization to multi-step GSP** We extend our one-step optimization to variable length sequence of actions in a straightforward manner by having a multi-step GSP $\pi_m$ model with a step-

wise forward consistency loss. The GSP $\pi_m$ maintains an internal recurrent memory of the system and outputs actions conditioned on current observation $x_t$, starting from $x_i$ to reach goal observation $x_T$. The forward consistency loss is computed at each time step, and jointly optimized with the action prediction loss over the whole trajectory. The final multi-step objective with feature space dynamics is as follows:

$$\min_{\theta_\pi, \theta_f, \theta_\phi} \sum_{t=i}^{t=T} \Big( \|\phi(x_{t+1}) - \tilde{\phi}(x_{t+1})\|_2^2 + \lambda\|\phi(x_{t+1}) - \hat{\phi}(x_{t+1})\|_2^2 + \mathcal{L}(a_t, \hat{a}_t)\Big) \quad (4)$$

$$\text{s.t.} \quad \tilde{\phi}(x_{t+1}) = f(\phi(x_t), a_t; \theta_f)$$
$$\hat{\phi}(x_{t+1}) = f(\phi(x_t), \hat{a}_t; \theta_f)$$
$$\hat{a}_t = \pi(\phi(x_t), \phi(x_T); \theta_\pi)$$

where $\phi(.)$ is represented by a CNN with parameters $\theta_\phi$. The number of steps taken by the multi-step GSP $\pi_m$ to reach the goal at inference is variable depending on the decision of goal recognizer; described in next subsection. Note that, in this objective, if $\phi$ is identity then the dynamics simply reduces to modeling in raw observation space. We analyze feature space prediction in *VizDoom* 3D navigation and stick to observation space in the rope manipulation and the office navigation tasks.

The multi-step *forward-consistent GSP* $\pi_m$ is implemented using a recurrent network which at every step takes as input the feature representation of the current ($\phi(x_t)$) state, goal ($\phi(x_T)$) states, action at the previous time step ($a_{t-1}$) and the internal hidden representation $h_{t-1}$ of the recurrent units and predicts $\hat{a}_t$. Note that inputting the previous action to GSP $\pi_m$ at each time step could be redundant given that hidden representation is already maintaining a history of the trajectory. Nonetheless, it is helpful to explicitly model this history. This formulation amounts to building an auto-regressive model of the joint action that estimates probability $P(a_t|x_1, a_1, ...a_{t-1}, x_t, x_g)$ at every time step. It is possible to further extend our forward-consistent GSP $\pi_m$ to build multi-step forward model, but we leave that direction of future work.

## 2.3 GOAL RECOGNIZER

We train a goal recognizer network to figure out if the current goal is reached and therefore allow the agent to take variable numbers of steps between goals. Goal recognition is especially critical when the agent has to transit through a sequence of intermediate goals, as is the case for visual imitation, as otherwise compounding error could quickly lead to divergence from the demonstration. This recognition is simple given knowledge of the true physical state, but difficult when working with visual observations. Aside from the usual challenges of visual recognition, the dependence of observations on the agent's own dynamics further complicates goal recognition, as the same goal can appear different while moving forward or turning during navigation.

We pose goal recognition as a binary classification problem that given an observation $x_i$ and the goal $x_g$ infers if $x_i$ is close to $x_g$ or not. Lacking expert supervision of goals, we draw goal observations at random from the agent's experience during exploration, since they are known to be feasible. For each such pseudo-goal, we consider observations that were only a few actions away to be positives (i.e., close to the goal) and the remaining observations that were more than a fixed number of actions (i.e., a margin) away as negatives. We trained the goal classifier using the standard cross-entropy loss. Like the skill policy, our goal recognizer is conditioned on the goal for generalization across goals. We found that training an independent goal recognition network consistently outperformed the alternative approach that augments the action space with a "stop" action. Making use of temporal proximity as supervision has also been explored for feature learning in the concurrent work of Sermanet et al. (2018).

## 2.4 ABLATIONS AND BASELINES

Our proposed formulation of GSP composed of following components: (a) recurrent variable-length skill policy network, (b) explicitly encoding previous action in the recurrence, (c) goal recognizer, (d) forward consistency loss function, and (w) learning forward dynamics in the feature space instead of raw observation space. We systematically ablate these components of forward-consistent GSP, to quantitatively review the importance of each component and then perform comparisons to the prior approaches that could be deployed for the task of visual imitation.

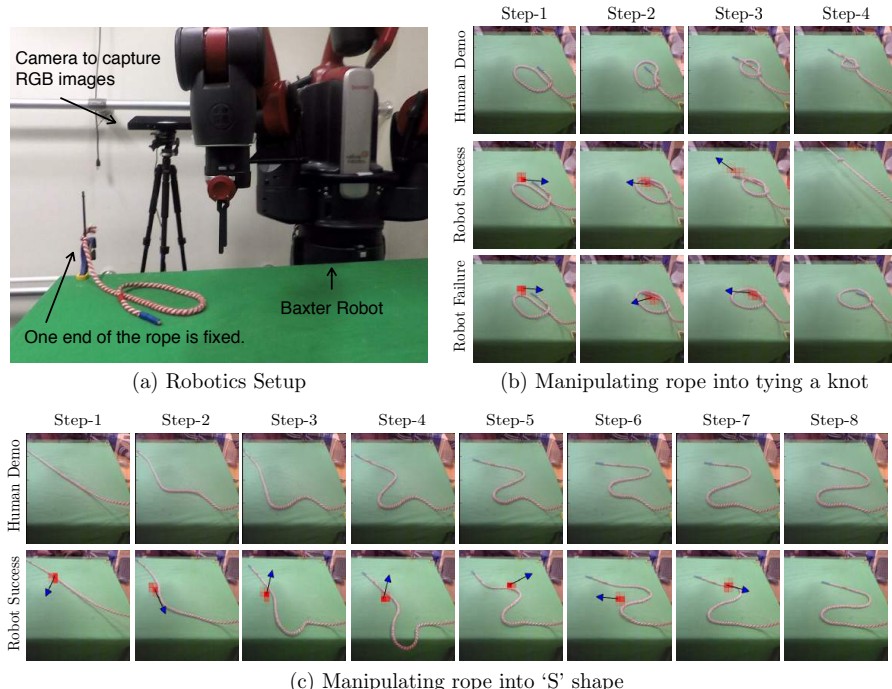

Figure 2: Qualitative visualization of results for rope manipulation task using Baxter robot. (a) Our robotics system setup. (b) The sequence of human demonstration images provided by the human during inference for the task of knot-tying (top row), and the sequences of observation states reached by the robot while imitating the given demonstration (bottom rows). (c) The sequence of human demonstration images and the ones reached by the robot for the task of manipulating rope into 'S' shape. Our agent is able to successfully imitate the demonstration.

The following methods will be evaluated and compared to in the subsequent experiments section: (1) Classical methods: In visual navigation, we attempted to compare against the state-of-the-art open source classical methods, namely, ORB-SLAM2 (Davison & Murray, 1998; Mur-Artal & Tardós, 2017) and Open-SFM (Mapillary, 2016). (2) Inverse Model: Nair et al. (2017) leverage vanilla inverse dynamics to follow demonstration in rope manipulation setup. We compare to their method in both visual navigation and manipulation. (3) GSP-NoPrevAction-NoFwdConst is the ablation of our recurrent GSP without previous action history and without forward consistency loss. (4) GSP-NoFwdConst refers to our recurrent GSP with previous action history, but without forward consistency objective. (5) GSP-FwdRegularizer refers to the model where forward prediction is only used to regularize the features of GSP but has no role to play in the loss function of predicted actions. The purpose of this variant is to particularly ablate the benefit of consistency loss function with respect to just having forward model as feature regularizer. (6) GSP refers to our complete method with all the components. We now discuss the experiments and evaluate these baselines.

## 3 EXPERIMENTS

We evaluate our model by testing its performance on: rope manipulation using Baxter robot, navigation of a wheeled robot in cluttered office environments, and simulated 3D navigation. The key requirements of a good skill policy are that it should generalize to unseen environments and new goals while staying robust to irrelevant distractors in the observations. For rope manipulation, we evaluate generalization by testing the ability of the robot to manipulate the rope into configurations such as knots that were not seen during random exploration. For navigation, both real-world and simulation, we check generalization by testing on a novel building/floor.

### 3.1 ROPE MANIPULATION

Manipulation of non-rigid and deformable objects, e.g., rope, is a challenging problem in robotics. Even humans learn complex rope manipulation such as tying knots, either by observing an expert

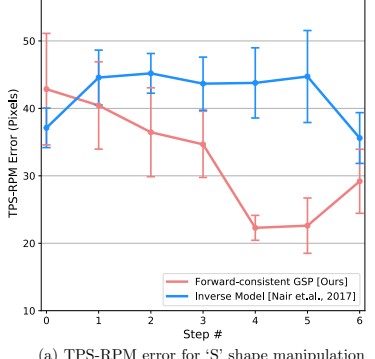

| Method | Success % |
|---|---|
| Inverse Model [Nair et.al. 2017] | $36\% \pm 9.6\%$ |
| Forward-regularized GSP | $44\% \pm 9.9\%$ |
| Forward-consistent GSP [Ours] | $\mathbf{60}\% \pm \mathbf{9.8}\%$ |

(a) TPS-RPM error for 'S' shape manipulation        (b) Success rate for Knot-tying

Figure 3: GSP trained using forward consistency loss significantly outperforms the baselines at the task of (a) manipulating rope into 'S' shape as measured by TPS-RPM error and (b) knot-tying where we report success rate with bootstrap standard deviation.

perform it or by receiving explicit instructions. We test whether our agent could manipulate ropes by simply observing a human perform it. We use the data collected by Nair et al. (2017), where a Baxter robot manipulated a rope kept on the table in front of it. During exploration, the robot interacts with the rope by using a pick and place primitive that chooses a random point on the rope and displaces it by a randomly chosen length and direction. This process is repeated a number of times to collect about 60K interaction pairs of the form $(x_t, a_t, x_{t+1})$ that are used to train the GSP.

During inference, our proposed approach is tasked to follow a visual demonstration provided by a human expert for manipulating the rope into a complex 'S' shape and tying a knot. Our agent, Baxter robot, only gets to observe the image sequence of intermediate states, as human manipulates the rope, without any access to the corresponding actions. Note that the knot shape is never encountered during the self-supervised data collection phase and therefore the learned GSP model would have to generalize to be able to follow the human demonstration. More details follow in the supplementary material, Section A.1.

**Metric**  The performance of the model is evaluated by measuring the non-rigid registration cost between the rope state achieved by the robot and the state demonstrated by the human at every step in the demonstration. The matching cost is measured using the thin plate spline robust point matching technique (TPS-RPM) described in (Chui & Rangarajan, 2003). While TPS-RPM provides a good metric for measuring performance for constructing the 'S' shape, it is not an appropriate metric for knots because the configuration of the rope in a knot is 3D due to intertwining of the rope, and it fails to find the correct point correspondences. We, therefore, use success rate as the metric in knot tying where the completion of a successful knot is judged by human verification.

**Visual Imitation**  Qualitative examples of our agent trying to manipulate rope are shown in Figure 2. We compare our approach to the baseline that deploys an inverse model which takes as input a pair of current and goal images to output the desired action to reach the goal (Nair et al., 2017). We re-implement the baseline and train in our setup for a fair comparison. To further ablate the importance of consistency loss, we compare to a baseline that just uses a forward model as a regularizer of features. The results in Figure 3 show that our method significantly outperforms the baseline at task of manipulating the rope in the 'S' shape and achieves a success rate of $60\%$ in comparison to $36\%$ achieved by the baseline.

## 3.2 NAVIGATION IN INDOOR OFFICE ENVIRONMENTS

A natural way to instruct a robot to move in an indoor office environment is to ask it to go near a certain location, such as a refrigerator or a someone's office. Instead of using language to command the robot, in this work, we communicate with the robot by either showing it a single image of the goal, or a sequence of images leading to faraway goals. In both scenarios, the robot is required to autonomously determine the motor commands for moving to the goal. We used TurtleBot2 for navigation using an onboard camera for sensing RGB images. For learning the GSP, an automated

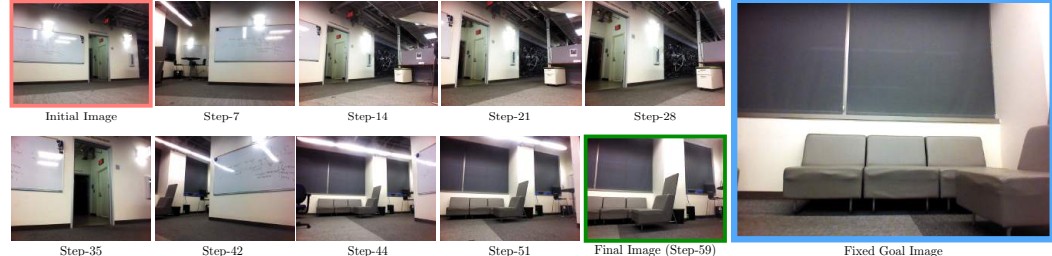

| Initial Image | Step-7 | Step-14 | Step-21 | Step-28 |
|---|---|---|---|---|
| Step-35 | Step-42 | Step-44 | Step-51 | Final Image (Step-59) | Fixed Goal Image |

Figure 4: Visualization of the TurtleBot trajectory to reach a goal image (right) from the initial image (top-left). Since the initial and goal image have no overlap, the robot first explores the environment by turning in place. Once it detects overlap between its current image and goal image (i.e. step 42 onward), it moves towards the goal. Note that we did not explicitly train the robot to explore and such exploratory behavior naturally emerged from the self-supervised learning.

| Model Name | Run Id-1 | Run Id-2 | Run Id-3 | Run Id-4 | Run Id-5 | Run Id-6 | Run Id-7 | Run Id-8 | Num Success |
|---|---|---|---|---|---|---|---|---|---|
| Random Search | Fail | Fail | Fail | Fail | Fail | Fail | Fail | Fail | 0 |
| Inverse Model [Nair et. al. 2017] | Fail | Fail | Fail | Fail | Fail | Fail | Fail | Fail | 0 |
| GSP-NoPrevAction-NoFwdConst | 39 steps | 34 steps | Fail | Fail | Fail | Fail | Fail | Fail | 2 |
| GSP-NoFwdConst | 22 steps | 22 steps | 39 steps | 48 steps | Fail | Fail | Fail | Fail | 4 |
| GSP (Ours) | 119 steps | 66 steps | 144 steps | 67 steps | 51 steps | Fail | 100 steps | Fail | 6 |

Table 1: Quantitative evaluation of various methods on the task of navigating using a *single image* of goal in an unseen environment. Each column represents a different run of our system for a different initial/goal image pair. Our full GSP model takes longer to reach the goal on average given a successful run but reaches the goal successfully at a much higher rate.

self-supervised scheme for data collection was devised that doesn't require human supervision. The robot collected a number of navigation trajectories from two floors of a academic building which in total contain 230K interactions data, i.e. $(x_t, a_t, x_{t+1})$. We then deployed the learned model on a separate floor of a building with substantially different textures and furniture layout for performing visual imitation at test time. The details of the robotic setup, data collection, and network architecture of GSP are described in supplementary material, Section A.2.

**1) Goal Finding** We first tested if the GSP learned by the TurtleBot can enable it to find its way to a goal that is within the same room from just a *single image* of the goal. To test the extrapolative generalization, we keep the Turtlebot approximately 20-30 steps away from the target location in a way that current and goal observations have no overlap as shown in Figure 4. We test the robot in an indoor office environment on a different floor that it has never encountered before. We judge the robot to be successful if it stops close to the goal and failure if it crashed into furniture or does not reach the goal within 200 steps. Since the initial and goal images have no overlap, classical techniques such as structure from motion that rely on feature matching cannot be used to infer the executed action. Therefore, in order to reach the goal, the robot must explore its surroundings. We find that our GSP model outperforms the baseline models in reaching the target location. Our model learns the exploratory behavior of rotating in place until it encounters an overlap between its current and goal image. Results are shown in Table 1 and videos are available at the website [1].

**2) Visual Imitation** In the previous paragraph, we saw that the robot can reach a goal that's within the same room. However, our agent is unable to reach far away goals such as in other rooms using just a single image. In such scenarios, an expert might communicate instructions like go to the door, turn right, go to the closest chair etc. Instead of language instruction, in our setup we provide a sequence of *landmark* images to convey the same high-level idea. These landmark images were captured from the robot's camera as the expert moved the robot from the start to a goal location. However, note that it is not necessary for the expert to control the robot to capture the images because we don't make use of the expert's actions, but only the images. Instead of providing the image after every action in the demonstration, we only provided every fifth image. The rationale behind this choice is that we want to sample the demonstration sparsely to minimize the agent's

---
[1]https://pathak22.github.io/zeroshot-imitation/

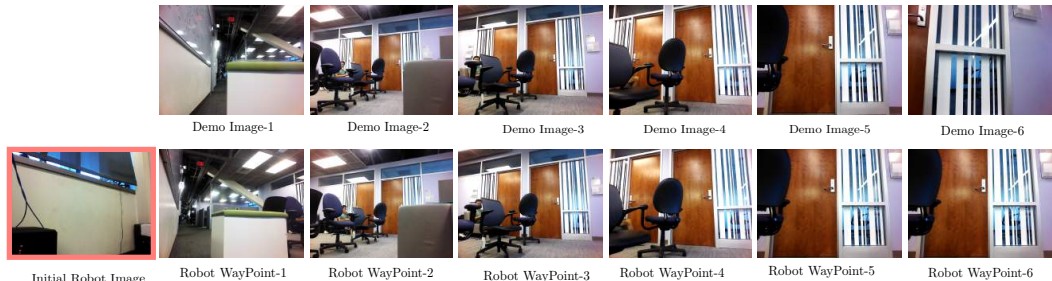

Figure 5: The performance of TurtleBot at following a visual demonstration given as a sequence of images (top row). The TurtleBot is positioned in a manner such that the first image in demonstration has no overlap with its current observation. Even under this condition the robot is able to move close to the first demo image (shown as Robot WayPoint-1) and then follow the provided demonstration until the end. This also exemplifies a failure case for classical methods; there are no possible key-point matches between WayPoint-1 and WayPoint-2, and the initial observation is even farther from WayPoint-1.

| Model Name | Maze Demonstration | | | Loop Demonstration | | |
|---|---|---|---|---|---|---|
| | Run-1 | Run-2 | Run-3 | Run-1 | Run-2 | Run-3 |
| SIFT | 10% | 5% | 15% | — | — | — |
| GSP-NoPrevAction-NoFwdConst | 60% | 70% | 100% | — | — | — |
| GSP-NoFwdConst | 65% | 90% | 100% | 0% | 0% | 0% |
| GSP (ours) | 100% | 60% | 100% | 0% | 100% | 100% |

Table 2: Quantitative evaluation of TurtleBot's performance at following visual demonstrations in two scenarios: maze and the loop. We report the % of landmarks reached by the agent across three runs of two different demonstrations. Results show that our method outperforms the baselines. Note that 3 more trials of the loop demonstration were tested under significantly different lighting conditions and neither model succeeded. Detailed results are available in the supplementary materials.

reliance on the expert. Such sub-sampling (as shown in Figure 5) provides an easy way to vary the complexity of the task.

We evaluate via multiple runs of two demonstrations, namely, maze demonstration where the robot is supposed to navigate through a maze-like path and perturbed loop demonstration, where the robot is supposed to make a complete loop as instructed by demonstration images. The loop demonstration is longer and more difficult than the maze. We start the agent from different starting locations and orientations with respect to that of demonstration. Each orientation is initialized such that no part of the demonstration's initial frame is visible. Results are shown in Table 2. When we sample every frame, our method and classical structure from motion can both be used to follow the demonstration. However, at sub-sampling rate of five, SIFT-based feature matching approaches did not work and ORBSLAM2 (Mur-Artal & Tardós, 2017) failed to generate a map, whereas our method was successful. Notice that providing sparse landmark images instead of dense video adds robustness to the visual imitation task. In particular, consider the scenario in which the environment has changed since the time the demonstration was recorded. By not requiring the agent to match every demonstration image frame-by-frame, it becomes less sensitive to changes in the environment.

### 3.3 3D NAVIGATION IN VIZDOOM

We have evaluated our approach on real-robot scenarios thus far. To further analyze the performance and robustness of our approach through large scale experiments, we setup the same navigation task as described in previous subsection in a simulated VizDoom environment. Our goal is to measure: (1) the robustness of each method with proper error bars, (2) the role of initial self-supervised data collection for performance on visual imitation, (3) the quantitative difference in modeling forward consistency loss in feature space in comparison to raw visual space.

| Model Name | Same Map, Same Texture | | Same Map, Diff Texture | | Diff Map, Diff Texture | |
|---|---|---|---|---|---|---|
| | Median % | Efficiency % | Median % | Efficiency % | Median % | Efficiency % |
| *Random Exploration for Data Collection*: | | | | | | |
| GSP-NoFwdConst | $63.2 \pm 5.7$ | $36.4 \pm 3.3$ | $32.2 \pm 0.7$ | $28.9 \pm 4.0$ | $34.5 \pm 0.6$ | $23.1 \pm 2.4$ |
| GSP (ours pixels) | $62.2 \pm 5.1$ | $43.0 \pm 2.6$ | $32.4 \pm 0.8$ | $30.9 \pm 2.9$ | $35.4 \pm 1.1$ | $29.3 \pm 3.9$ |
| GSP (ours features) | $68.9 \pm 6.9$ | $53.9 \pm 4.0$ | $32.4 \pm 0.7$ | $47.4 \pm 7.6$ | $39.1 \pm 2.0$ | $30.4 \pm 2.5$ |
| *Curiosity-driven Exploration for Data Collection*: | | | | | | |
| GSP-NoFwdConst | $78.2 \pm 2.3$ | $63.0 \pm 4.3$ | $43.2 \pm 2.6$ | $33.9 \pm 3.0$ | $40.2 \pm 4.0$ | $27.3 \pm 1.9$ |
| GSP-FwdRegularizer | $78.4 \pm 3.4$ | $59.8 \pm 4.1$ | $50.6 \pm 4.7$ | $30.9 \pm 3.0$ | $37.9 \pm 1.1$ | $28.9 \pm 1.7$ |
| GSP (ours pixels) | $78.2 \pm 3.4$ | $65.2 \pm 4.2$ | $47.1 \pm 4.7$ | $32.4 \pm 3.0$ | $44.8 \pm 4.0$ | $29.5 \pm 1.9$ |
| GSP (ours features) | $78.2 \pm 4.6$ | $67.0 \pm 3.3$ | $49.4 \pm 4.8$ | $26.9 \pm 1.5$ | $47.1 \pm 3.0$ | $24.1 \pm 1.7$ |

Table 3: Quantitative evaluation of our proposed GSP and the baseline models at following visual demonstrations in VizDoom 3D Navigation. Medians and 95% confidence intervals are reported for demonstration completion and efficiency over 50 seeds and 5 human paths per environment type.

In VizDoom, we collect data by deploying two types of exploration methods: random exploration and curiosity-driven exploration (Pathak et al., 2017). The hypothesis is that if the initial data collected by the robot is driven by a better strategy than just random, this should eventually help the agent follow long demonstrations better. Our environment consists of 2 maps in total. We train on one map with 5 different starting positions for collecting exploration data. For validation, we collect 5 human demonstrations in a map with the same layout as in training but with different textures. For zero-shot generalization, we collect 5 human demonstrations in a novel map layout with novel textures. Exact details for data collection and training setup are in the supplementary, Section A.3.

**Metric** We report the median of maximum distance reached by the robot in following the given sequence of demonstration images. The maximum distance reached is the distance of farthest landmark point that the agent reaches contiguously, i.e., without missing any intermediate landmarks. Measuring the farthest landmark reached does not capture how efficiently it is reached. Hence, we further measure efficiency of the agent as the ratio of number of steps taken by the agent to reach farthest contiguous landmark with respect to the number of steps shown in human demonstrations.

**Visual Imitation** The task here is same as the one in real robot navigation where the agent is shown a sparse sequence of images to imitate. The results are in Table 3. We found that the exploration data collected via curiosity significantly improves the final imitation performance across all methods including the baselines with respect to random exploration. Our baseline GSP model with a forward regularizer instead of consistency loss ends up overfitting to the training layout. In contrast, our forward-consistent GSP model outperforms other methods in generalizing to new map with novel textures. This indicates that the forward consistency is possibly doing more than just regularizing the policy features. Training forward consistency loss in feature space further enhances the generalization even when both pixel and feature space models perform similarly on training environment.

## 4 RELATED WORK

Our work is closely related to imitation learning, but we address a different problem statement that gives less supervision and requires generalization across tasks during inference.

**Imitation Learning** The two main threads of imitation learning are behavioral cloning (Argall et al., 2009; Pomerleau, 1989), which directly supervises the mapping of states to actions, and inverse reinforcement learning (Abbeel & Ng, 2004; Ho & Ermon, 2016; Levine et al., 2016; Ng & Russell, 2000; Ziebart et al., 2008), which recovers a reward function that makes the demonstration optimal (or nearly optimal). Inverse RL is most commonly achieved with state-actions, and is difficult to extend to fitting the reward to observations alone, though in principle state occupancy could be sufficient. Recent work in imitation learning (Duan et al., 2017; Finn et al., 2017; Gupta et al., 2017) can generalize to novel goals, but require a wealth of demonstrations comprised of expert state-actions for learning. Our approach does not require expert actions at all.

**Visual Demonstration** The common scenario in LfD is to assume full knowledge of expert states and actions during demonstrations, but several papers have focused on relaxing this supervision to

visual observations alone. Nair et al. (2017) observe a sequence of images from the expert demonstration for performing rope manipulations. Sermanet et al. (2017; 2018) imitate humans with robots by self-supervised learning but require expert supervision at training time. Third person imitation learning (Stadie et al., 2017) and the concurrent work of imitation-from-observation (Liu et al., 2018) learn to translate expert observations into agent observations such that they can do policy optimization to minimize the distance between the agent trajectory and the translated demonstration, but they require demonstrations for learning. Visual servoing is a standard problem in robotics (Koichi & Tom, 1993) that seeks to take actions that align the agent's observation with a target configuration of carefully-designed visual features (Wilson et al., 1996; Yoshimi & Allen, 1994) or raw pixel intensities (Caron et al., 2013). Classical methods rely on fixed features or policies, but more recently end-to-end learning has improved results (Lampe & Riedmiller, 2013; Lee et al., 2017).

**Forward/Inverse Dynamics and Consistency**  Numerous prior works, such as Ebert et al. (2017); Oh et al. (2015); Watter et al. (2015), have learned forward dynamics model for planning actions. The works of Agrawal et al. (2016); Jordan & Rumelhart (1992); Pathak et al. (2017); Wolpert et al. (1995) jointly learn forward and inverse dynamics model but do not optimize for consistency between the forward and inverse dynamics. We empirically show that learning models by our forward consistency loss significantly improves task performance. Enforcing consistency as a meta-supervision has also been successful in finding visual correspondences (Zhou et al., 2016) or unpaired image translations (Zhu et al., 2017).

**Goal Conditioning**  By parameterizing the value or policy function with a goal, an agent can learn and do multiple tasks. The idea of learning goal-conditioned policies has been explored in (Agrawal et al., 2016; Andrychowicz et al., 2017; Nair et al., 2017; Schaul et al., 2015). Similarly to hindsight experience replay (Andrychowicz et al., 2017) we draw goals from experience, but our policy optimization has better sample efficiency through supervised learning and dynamics modeling instead of reinforcement learning. Moreover, we work from high-dimensional visual inputs instead of knowledge of the true states and do not make use of a task reward during training. In our setting, all of the expert goals are followed zero-shot since they are only revealed after learning.

## 5  DISCUSSION

In this work, we presented a method for imitating expert demonstrations from visual observations alone. In contrast to most work in imitation learning, we never require access to expert actions. The key idea is to learn a GSP using data collected by self-supervised exploration. However, this limits the quality of the learned GSP as per the exploration data. For instance, we deploy random exploration on our real-world navigation robot, which means that it would almost never follow trajectories that go between rooms. Consequently, the learned GSP is unable to navigate towards a goal image taken in another room without requiring intermediate sub-goals. Pathak et al. (2017) show that the agent learns to move along corridors and transition between rooms purely driven by curiosity in Viz-Doom. Training GSP on such a structured data could equip the agent with more interesting search behaviors, e.g., going across rooms to find a goal. In general, using better methods of exploration for training the GSP could be a fruitful direction toward generalizing zeroshot imitation.

One limitation of our approach is that we require first-person view demonstrations. Extension to third-person demonstrations (Liu et al., 2018; Stadie et al., 2017) would make the method applicable in more general scenarios. Another limitation is that, in the current framework, it is implicitly assumed that the statistics of visual observations when the expert demonstrates the task and the agent follows it are similar. For e.g., when the expert performs a demonstration in one setting, say in daylight and the agent needs to imitate say in the evening, the change in the lighting conditions might result in worse performance. Making the GSP robust to such nuisance changes or other changes in environment by domain adaptation would be necessary to scale the method to practical problems. Another thing to note is that, in the current framework, we do not learn from expert demonstrations, but simply imitate them. It would be interesting to investigate ways for an agent to learn from the expert to bias its exploration to more useful parts of the environment.

While we used a sequence of images to provide a demonstration, our work makes no image-specific assumptions and can be extended to using formal language for communicating goals. For instance, after training the GSP, instead of transforming an image into features $\phi$ as described in section 2.2, one could possibly learn a mapping to transform language instructions into this feature space.

ACKNOWLEDGMENTS

We would like to thank members of BAIR for fruitful discussions and comments. This work was supported in part by DARPA; NSF IIS-1212798, IIS-1427425, IIS-1536003, Berkeley DeepDrive, and an equipment grant from NVIDIA and the Valrhona Reinforcement Learning Fellowship. DP is supported by NVIDIA and Snapchat's graduate fellowships.

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

# A  SUPPLEMENTARY MATERIAL

We evaluated our proposed approach across number of environments and tasks. In this section, we provide additional details about the experimental task setup and hyperparameters.

## A.1  ROPE MANIPUATION

**Robotic Setup**  Our setup of Baxter robot for rope manipulation task follows the one described in Nair et al. (2017). We re-use the data that is collected by a Baxter robot interacting with a rope kept on a table in front of it in a self-supervised manner, and consists of approximately 60K interaction pairs.

**Implementation Details**  The base architecture for all the methods consists of a pre-trained AlexNet, whose features are fed into a skill policy network that predicts the location of grasp, direction of displacement, and the magnitude of displacement. For the forward regularizer baseline, a forward model is trained to jointly regularize the AlexNet features along with the skill policy network with loss weight of forward model set to 0.1. For our proposed forward-consistent GSP, a forward consistency loss is then applied to the actions predicted by the skill policy network. The forward consistency loss weight is set to 0.1. Since this is a fully observed setup, we did not use recurrence in any of the skill policy networks. All the models are optimized using Adam (Kingma & Ba, 2015) with a learning rate of $1e-4$. For the first 40K iterations, the AlexNet weights were frozen, and then fine-tuned jointly with the later layers.

## A.2  NAVIGATION IN INDOOR OFFICE ENVIRONMENTS

**Robotic Setup**  We used the TurtleBot2 robot comprising of a wheeled Kobuki base and an Orbbec Astra camera for capturing RGB images for all our experiments. The robot's action space had four discrete actions: move forward, turn left, turn right, and stand still (i.e., no-op). The forward action is approximately 10cm forward translation and the turning actions are approximately 14-18 degrees of rotation. These numbers vary due to the use of velocity control. A powerful on-board laptop was used to process the images and infer the motor commands. Several modifications were made to the default TurtleBot setup: the base's batteries were replaced with longer lasting ones, and the default NVIDIA Jetson TK1 embedded board was replaced with a more powerful GigaByte Aero laptop and an accompanying portable charging power bank.

**Self-supervised Data Collection**  We devised an automated self-supervised scheme for data collection which does not require any human supervision. In our scheme, the robot first samples one out of four actions and then the number of times to repeat the selected action (i.e. action repeat). The no-op action is sampled with probability 0.05 and the other three actions are sampled with equal probability. In case the no-op action is chosen, an action repeat of $\{1, 2\}$ steps is uniformly sampled. In case of other actions, an action repeat of 1-5 steps is randomly and uniformly chosen. The robot autonomously repeated this process and collected 230K interactions from two floors of an academic building. If the robot crashes into an object, it performs a reset maneuver by first moving backwards and then turning right/left by a uniformly sampled angle between 90-270 degrees. A separate floor of the building with substantially different furniture layout and visual textures is then used for testing the learned model.

**Implementation Details**  The data collected by self-supervised exploration is then used to train our *recurrent forward-consistent GSP*. The base architecture of our model is an ImageNet pre-trained ResNet-50 (He et al., 2016) network. Input are the images and output are the actions of robot. The forward consistency model is first pre-trained and then fine-tuned together end-to-end with the GSP. The loss weight of the forward model is 0.1, and the objective is minimized using Adam (Kingma & Ba, 2015) with learning rate of $5e-4$.

## A.3  3D NAVIGATION IN VIZDOOM

**Self-supervised Data Collection**  Our environment consists of two map. One map is used for training and validation, with different textures for validation. Second map has different textures than training and validation and is used for generalization experiments. For both curiosity and random exploration, we collect a total of 1.5 million frames each with action repeat of 4 collected in the

| Model Name | Maze Runs - Optimal Steps: 100 | | | Loop Runs - Optimal Steps: 85 | | |
|---|---|---|---|---|---|---|
| | Run-1 | Run-2 | Run-3 | Run-1 | Run-2 | Run-3 |
| SIFT | 2/20 (10) | 1/20 (9) | 3/20 (38) | — | — | — |
| GSP-NoPrevAction-NoFwdConst | 12/20 (109) | 14/20 (184) | 20/20 (263) | — | — | — |
| GSP-NoFwdConst | 13/20 (147) | 18/20 (325) | 20/20 (166) | 0/17 (0) | 0/17 (0) | 0/17 (0) |
| GSP (ours) | 20/20 (353) | 12/20 (194) | 20/20 (168) | 0/17 (0) | 17/17 (243) | 17/17 (165) |

Table 4: Quantitative evaluation of TurtleBot's performance at following visual demonstrations in two conditions: maze and the loop. The fraction denotes how many landmarks it reaches out of the total number of landmarks in the full demonstration. The bracketed number represents the number of actions the agent took to reach its farthest landmark.

| Model Name | Same Map, Same Texture | | Same Map, Diff Texture | | Diff Map, Diff Texture | |
|---|---|---|---|---|---|---|
| | Mean % | Efficiency % | Mean % | Efficiency % | Mean % | Efficiency % |
| *Random Exploration for Data Collection*: | | | | | | |
| GSP-NoFwdConst | $61.8 \pm 0.9$ | $60.4 \pm 2.1$ | $37.6 \pm 0.7$ | $68.6 \pm 2.5$ | $42.2 \pm 0.8$ | $50.6 \pm 1.9$ |
| GSP (ours pixels) | $61.0 \pm 1.0$ | $68.0 \pm 2.2$ | $38.1 \pm 0.7$ | $69.1 \pm 2.5$ | $40.3 \pm 0.9$ | $64.2 \pm 2.3$ |
| GSP (ours features) | $62.0 \pm 1.0$ | $75.8 \pm 2.5$ | $37.0 \pm 0.7$ | $87.1 \pm 2.8$ | $48.7 \pm 0.9$ | $52.5 \pm 1.8$ |
| *Curiosity-driven Exploration for Data Collection*: | | | | | | |
| GSP-NoFwdConst | $70.7 \pm 0.9$ | $66.9 \pm 1.4$ | $49.8 \pm 0.8$ | $55.8 \pm 2.2$ | $51.2 \pm 1.0$ | $39.5 \pm 1.3$ |
| GSP-FwdRegularizer | $70.6 \pm 0.9$ | $67.9 \pm 1.6$ | $51.9 \pm 0.8$ | $49.3 \pm 1.6$ | $48.3 \pm 1.0$ | $49.3 \pm 1.8$ |
| GSP (ours pixels) | $71.0 \pm 0.9$ | $73.1 \pm 2.7$ | $53.3 \pm 0.9$ | $53.4 \pm 2.0$ | $52.2 \pm 1.0$ | $44.0 \pm 1.5$ |
| GSP (ours features) | $68.8 \pm 1.0$ | $72.0 \pm 1.7$ | $53.2 \pm 0.8$ | $53.0 \pm 2.3$ | $52.8 \pm 0.9$ | $37.7 \pm 1.3$ |

Table 5: Quantitative evaluation of our proposed GSP and the baseline models at following visual demonstrations in VizDoom 3D Navigation. Means and standard errors are reported for demonstration completion and efficiency over 50 seeds and 5 human paths per environment type.

standard `DoomMyWayHome` map used for training in Pathak et al. (2017). $\sim \frac{2}{3}$ of the data comes from random-room resets, and $\sim \frac{1}{3}$ of the data comes from a fixed-room reset (i.e, room number 10). The curiosity policy was half sampled and half greedy with the exact split being 40% greedy policy random-room reset, 25% sample policy random-room reset, 25% sample policy fixed-room reset, and 10% greedy policy fixed-room reset.

For each scenario, we collect 5 human demonstrations each and give every 10th frame as input to the agent for the task of visual imitation. For each human path, we evaluate on 50 different seeds where the agent starts with a uniformly sampled orientation. We then get the median across 250 (50x5) total runs for each type of environment and report median of the percentage of the human path reached by the agent and how soon it got to that point relative to the human.

In the main paper, we report median accuracy and the confidence interval for median [2]. Since the initial position of the agent is randomized in orientation compared to the one in visual demonstration, the mean results suffer from high variance due to outliers. Hence, median accuracy results in a more reliable metric. However, we report mean results in Table 5 for the completion.

**Implementation Details** All models were trained with batch size 64, Adam Solver with 1e-4 learning rate, and landmark slices uniformly sampled between 5 to 15 action steps for each batch. The observations are 42x42 resolution, grayscale images with only one-time channel both for goal and current state. All models used the same goal recognizer that was trained on the curiosity data. For selecting the hyper-parameters in forward regularizer, pixel-based forward consistency, and feature-based forward consistency models, we selected the best loss coefficient among $\{0.01, 0.05, 0.1\}$ that achieved the highest median completion on our validation environment which consisted of the training maps with novel textures.

---

[2]Formula for computing median confidence intervals: `http://www.ucl.ac.uk/ich/short-courses-events/about-stats-courses/stats-rm/Chapter_8_Content/confidence_interval_single_median`

