# OpenReview forum: "Zero-Shot Visual Imitation"
_ICLR.cc/2018/Conference — Accept (Oral)_

### Official Review · AnonReviewer3 · 2017-11-27
**A good approach with some open questions on related work, scalability, and robustness**

**Rating:** 8
**Confidence:** 4

**Review:**

The authors propose an approach for zero-shot visual learning. The robot learns inverse and forward models through autonomous exploration. The robot then uses the learned parametric skill functions to reach goal states (images) provided by the demonstrator. The “zero-shot” refers to the fact that all learning is performed before the human defines the task. The proposed method was evaluated on a mobile indoor navigation task and a knot tying task.

The proposed approach is well founded and the experimental evaluations are promising. The paper is well written and easy to follow.

I was expecting the authors to mention “goal emulation” and “distal teacher learning” in their related work. These topics seem sufficiently related to the proposed approach that the authors should include them in their related work section, and explain the similarities and differences.

Learning both inverse and forward models is very effective. How well does the framework scale to more complex scenarios, e.g., multiple types of manipulation together? Do you have any intuition for what kind of features or information the networks are capturing? For the mobile robot, is the robot learning some form of traversability affordances, e.g., recognizing actions for crossings, corners, and obstacles? The authors should consider a test where the robot remains stationary with a fixed goal, but obstacles are move around it to  see how it affects the selected action distributions.

How much can change between the goal images and the environment before the system fails? In the videos, it seems that the people and chairs are always in the same place. I could imagine a network learning to ignore features of objects that tend to wander over time. The authors should consider exploring and discussing the effects of adding/moving/removing objects on the performance.

I am very happy to see experimental evaluations on real robots, and even in two different application domains. Including videos of failure cases is also appreciated. The evaluation with the sequence of checkpoints was created by using every fifth image. How does the performance change with the number of frames between checkpoints? In the videos, it seems like the robot could get a slightly better view if it took another couple of steps. I assume this is an artifact of the way the goal recognizer is trained. For the videos, it may be useful to indicate when the goal is detected, and then let it run a couple more steps and stop for a second. It is difficult to compare the goal image and the video otherwise.

---

> ### Author Response · Authors · 2017-12-26
> **Response to AnonReviewer3**
>
> We are glad that the reviewer found the proposed approach well founded and the evaluations promising. We thank the reviewer for the constructive feedback and address the concerns in detail below.
>
> R3: "... mention goal emulation and distal teacher learning in their related work"
> => We thank the reviewer for bringing this to our notice. We will include these in the related work and highlight the differences.
>
> R3: "How well does the framework scale ... any intuition for what kind of features or information the networks are capturing?"
> => One interesting insight from the training process is that forward consistency loss improves the accuracy of inverse model even if the predicted image quality of forward model is not pixel accurate. One explanation is that the forward model is not required to be pixel-accurate till the time gradients are regularized, which leads to better performance.
>
> Regarding what information is being captured -- intuitively inverse models only represent information that is required to predict the action. We will include nearest neighbor visualization in the final revision of the paper to provide further insights into the nature of the learned representations.
>
> As far as complex manipulation tasks are concerned, we have shown results for manipulating a rope into  ‘S’ shape and tying into a knot. For more complex tasks and scaling to high dimensional spaces, we believe that instead of random exploration, more structured intrinsic motivation driven exploration, e.g. pseudo-count (Bellemare et al, 2016) or learning progress (Schmidhuber, 1991) or curiosity (Pathak et al, 2017) will play key role. We discuss this briefly in the Section-5 of the paper and will elaborate in more detail in the next revision.
>
> R3: "... authors should consider a test where the robot remains stationary with a fixed goal, but obstacles are move around ... discussing the effects ... on the performance"
> => Thank you for the suggestion. This is an interesting experiment and we will look into it. In our preliminary experiments, we found that the turtlebot is robust to dynamic obstacles as long as goal image does not change significantly.
>
> R3: "... the performance change with the number of frames between checkpoints?"
> => Generally speaking, as the number of frames between checkpoints increases, the performance deteriorates, but quite gracefully. To quantify this effect, we conducted an experiment where in the navigation setup the robot was only shown the goal image (~approximately 15-25 steps away from the current image on an optimal route). In this scenario, our robot exhibited  goal searching behavior until the goal comes in field of view, followed by goal directed navigation. Our method significantly outperformed other baselines as reported in Table-1.
>
> R3: "... it seems like the robot could get a slightly better view if it took another couple of steps ... useful to indicate when the goal is detected in video"
> => Yes, it is indeed the artifact of goal recognizer training because it was trained with some stochasticity around the arbitrarily selected states from collected exploration data.
>
> Thanks for the suggestions on making the result videos more comprehensible on how to analyze goal recognizer stop model. We will include qualitative analysis of the goal recognizer stop model in the supplementary materials of our final revision.

---

### Official Review · AnonReviewer2 · 2017-11-28
**Interesting paper on imitation of (relatively simple) tasks enhanced by previous self-exploration**

**Rating:** 8
**Confidence:** 3

**Review:**

Summary:
The authors present a paper about imitation of a task presented just during inference, where the learning is performed in a completely self-supervised manner.
During training, the agent explores by itself related (but different) tasks, learning a) how actions affect the world state, b) which action to perform given the previous action and the world state, and c) when to stop performing actions. This learning is done without any supervision, with a loss that tries to predict actions which result in the state achieved through self exploration (forward consistency loss).
During testing, the robot is presented with a sequence of goals in a related but different task. Experiments show that the system achieves a better performance than different subparts of the system (through an ablation study), state of the art and common open source systems.

Positive aspects:
The paper is well written and clear to understand. Since this is not my main area of research I cannot judge its originality in a completely fair way, but it is original AFAIK. The idea of learning the basic relations between actions and state through self exploration is definitely interesting.
This line of work is specially relevant since it attacks one of the main bottlenecks in learning complex tasks, which is the amount of supervised examples.
The experiments show clearly that a) the components of the proposed pipeline are important since they outperform ablated versions of it and b) the system is better than previous work in those tasks

Negative aspects:
My main criticism to the paper is that the task learning achieved through self exploration seems relatively shallow. From the navigation task, it seems like the system mainly learns a discover behavior that is better than random motion. It definitely does not seem able to learn higher level concepts like certain scenes being more likely to be close to each other than others (e.g. it is likely to find an oven in the same room as a kitchen sink but not in a toilet). It is not clear whether this is achievable by the current system even with more training data.
Another aspect that worries me about the system is how it can be extended to higher dimensional action spaces. Extending control laws through self-exploration under random disturbances has been studied in character control (e.g. "Domain of Attraction Expansion for Physics-based Character Control" by Borno et al.), but the dimensionality of the problem makes this exploration very expensive (even for short time frames, and even in simulation). I wonder if the presented ideas won't suffer from the same curse of dimensionality.
In terms of experiments, it is shown that the system is more effective than others but not so much *how* it achieves this efficiency. It would be good to show whether part of its efficiency comes from effective image-guided navigation: does a partial image match entail with targetted navigation (e.g. matches in the right side of the image make the robot turn right)?
A couple more specific comments:
- I think that dealing with multimodal distributions of actions with the forward consistency loss is effective for achieving the goal, but not necessarily good for modeling multimodality. Isn't it possible that the agent learns only one way of achieving such goal?
 - It is not clear how the authors achieve to avoid the problem of starting from scratch by "pre-train the forward model and PSF separately by blocking gradient flow". Isn't it still challenging to update them independently, given that at the beginning both components are probably not very accurate?


Conclusion:
I think the paper presents an interesting idea which should be exposed to the community. The paper is easy to read and its experiments show the effectiveness of the method. The relevance of the method to achieve a deeper sense of learning and performing more complex tasks is however unclear to me.

---

> ### Author Response · Authors · 2017-12-26
> **Response to AnonReviewer2 [part 1/2]**
>
> We thank the reviewer for the constructive feedback and are glad that the reviewer found the idea original and general direction as specially relevant and interesting. We address the concerns in detail below.
>
> R2: "... learns a discover behavior that is better than random motion ... not clear whether this is achievable by the current system even with more training data."
> => In the current setup of navigation, the exploration is random, and the robot learns the skills of avoiding walls, moving in free spaces and turning around to find the target image, etc. Note that we report performance in a setup when the robot is dropped in entirely new environments, showing that the learned skills generalize. In essence, what we have shown is that is possible to distill exploration data into generalizable skills that can be used to reach target images.
>
> The complexity of the skills learned by our robot inevitably depends on the interaction data it collects via its exploration. We agree that random exploration is insufficient for learning more higher-level concepts. There are many works in the literature, such as pseudo-count (Bellemare et al, 2016) or learning progress (Schmidhuber, 1991) or curiosity (Pathak et al, 2017), that have proposed efficient exploration schemes. In the future, we plan to incorporate these exploration policies to collect data and train PSF (parameterized skill function) with the help of this data. The intuition is that with a good exploration policy, the kitchen sink and the oven will be closer to each other as compared to the toilet in robot’s roll outs. The PSF learned using this roll out data is therefore likely to learn these relationships implicitly. Our goal in this work is not to propose an exploration policy, but a mechanism that can make use of exploration data to learn a skill function to achieve desired goals. Our initial experiments in simulation suggest that performance of goal reaching improves when data is collected using a non-random exploration policy.
>
> R2: "... how it can be extended to higher dimensional action spaces ... curse of dimensionality."
> => This is a great point. In our opinion there are two mechanisms: (a) discovering a low-dimensional embedding of the action space (say using motor babbling) and controlling in this space; (b) a more general mechanism is to make use of structured exploration mechanisms that has a rich literature. Some works incentivize the agent to visit previously unseen states, e.g., pseudo-count (Bellemare et al, 2016), other incentivize the agent to take actions that lead to high-prediction error (Pathak et al, 2017) or measure learning progress (Schmidhuber, 1991). Ours proposed forward consistent way of learning PSF is agnostic to how exploration data is collected. A PSF learned over the data obtained using any of these structured exploration mechanisms has the potential to scale to high dimensional spaces. We acknowledge this issue in the Section-5 of paper and will make it clearer in the final revision.
>
> R2: "... it is shown that the system is more effective than others but not so much *how* it achieves this efficiency ... partial image match"
> => There are two perspectives: (a) the quantitative view that can be used to systematically ablate the model to understand what components of the formulation are most critical and (b) a more qualitative and intuitive perspective, as suggested by you, that analyzes whether our robot learns to make a right/left turns more accurately when trained with forward consistency loss.
>
> For (a), as an ablation, we trained the inverse model with the forward loss as a regularizer. In this forward regularizer setup, the forward model and inverse model are trained jointly and share the visual features. The difference from our proposed approach is that inverse model is trained with action prediction loss and does not receive gradients through the forward model. In case of knot tying task, the forward regularizer achieves 44% accuracy which is above the baseline (36%) but well below our proposed approach (60%). This result shows that the forward model is not merely acting as a regularizer, but optimizing the inverse model through the forward model is potentially critical to addressing the multi-modality issue. We will add these numbers in the final version of the paper.
>
> For (b), when images have little overlap we found that classical solutions based on SIFT failed due to lack of keypoint matches. However, the inverse model and our approach are able to make left and right turns depending on whether the right or left part of the current image is visible in the goal image. We will include quantitative evaluation for this analysis in the final version of the paper.

---

> > ### Author Response · Authors · 2017-12-26
> > **Response to AnonReviewer2 [part 2/2]**
> >
> >
> > R2: "... forward consistency loss is effective for achieving the goal, but not necessarily good for modeling multimodality ... learns only one way of achieving such goal?"
> > => Yes, you are correct. We do-not directly model the multimodal distribution of the action, but we address the instability issues of gradient based learning due to multimodality. In an attempt to match the multiple ground-truth targets for the same input, the predictions will oscillate, which in turn will make the gradient of the loss function with respect to neural network parameters also oscillate. The purpose of forward consistency loss is to mitigate gradient oscillation, i.e. -- it stabilizes the learning process by ensuring that network is not penalized for outputting a different action than ground truth as long as its predicted action has the same effect as ground truth one.
> >
> > Learning all possible ways of achieving a goal is slightly different question. In theory, it could be dealt by incorporating a stochastic sampling layer in the neural network in addition to forward consistency and is an interesting direction for future research.
> >
> > R2: "... how the authors avoid the problem of starting from scratch by pre-train the forward model and PSF separately ... Isn't it still challenging"
> > => Training the PSF through forward consistency loss is a challenging problem because the learning of inverse model PSF depends on how good the forward model is. This learned forward model will not be very accurate in the beginning of training, thus making the gradients noisy. Therefore, we first pretrained inverse and forward model independently until convergence, and then fine-tune them jointly with consistency loss. Empirically, we found that such a pre-training followed by joint fine-tuning to be more stable than joint fine-tuning from scratch.

---

### Official Review · AnonReviewer1 · 2017-11-28
**Why is this not simply doing RL in the real world and then using the learnt policy (with no exploration)?**

**Rating:** 7
**Confidence:** 5

**Review:**

One of the main problems with imitation learning in general is the expense of expert demonstration. The authors here propose a method for sidestepping this issue by using the random exploration of an agent to learn generalizable skills which can then be applied without any specific pretraining on any new task.

The proposed method has at its core a method for learning a parametric skill function (PSF) that takes as input a description of the initial state, goal state, parameters of the skill and outputs a sequence of actions (could be of varying length) which take the agent from initial state to goal state.

The skill function uses a RNN as function approximator and minimizes the sum of two losses i.e. the state mismatch loss over the trajectory (using an explicitly learnt forward model) and the action mismatch loss (using a model-free action prediction module) . This is hard to do in practice due to jointly learning both the forward model as well as the state mismatches. So first they are separately learnt and then fine-tuned together.

In order to decide when to stop, an independent goal detector is trained which was found to be better than adding a 'goal-reached' action to the PSF.

Experiments on two domains are presented. 1. Visual navigation where images of start and goal states are given as input. 2. Robotic knot-tying with a loose rope where visual input of the initial and final rope states are given as input.

Comments:

- In the visual navigation task no numbers are presented on the comparison to slam-based techniques used as baselines although it is mentioned that it will be revisited.

- In the rope knot-tying task no slam-based or other classical baselines are mentioned.

- My main concern is that I am really trying to place this paper with respect to doing reinforcement learning first (either in simulation or in the real world itself, on-policy or off-policy) and then just using the learnt policy on test tasks. Or in other words why should we call this zero-shot imitation instead of simply reinforcement learnt policy being learnt and then used. The nice part of doing RL is that it provides ways of actively controlling the exploration. See this pretty relevant paper which attempts the same task and also claims to have the target state generalization ability.

Target-driven Visual Navigation in Indoor Scenes using Deep Reinforcement Learning by Zhu et al.

I am genuinely curious and would love the authors' comments on this. It should help make it clearer in the paper as well.

Update:

After evaluating the response from the authors and ensuing discussion as well as the other reviews and their corresponding discussion, I am revising my rating for this paper up. This will be an interesting paper to have at the conference and will spur more ideas and follow-on work.

---

> ### Author Response · Authors · 2017-12-26
> **Response to AnonReviewer1 [new baselines and discussion]**
>
> We thank you for the constructive feedback and address the concerns in detail below.
>
> R1: "In visual navigation ... no numbers for slam-based techniques ... will be revisited."
> => When the imitator is shown a dense demonstration sequence (i.e., every frame), it is possible to use SIFT features to estimate the odometry and guide the robot. However, in the more interesting scenario of the imitator being shown a sparser demonstration (i.e., every 5th frame), SIFT matching fails. The major reason for failure is the wide baseline, which on many occasions leads to little overlap between the current image and the next image in the demonstration. We will add SIFT numbers to the final revision.
>
> We also tried state-of-the-art open source methods: OpenSFM and ORB SLAM2. These methods were unable to generate the map with every 5th image of a demonstration sequence. The other possibility was to let the robot explore randomly and build a map. However, with random exploration the robot tends to wander off and is not focussed on constructing a map of the part of the environment from where the demonstration sequences were taken. This leads to failure in following the demonstration.
>
> R1: "In rope knot-tying task, no other classical baselines are mentioned"
> => Thanks for bringing this point up. The analog of SLAM in rope manipulation is to perform non-linear alignment of rope between the current and target image and use this alignment to select the action. TPS-RPM is a well-known method to align deformable objects such as ropes and is described in Section-4.2. TPS-RPM based method was compared in Nair et al. in which their inverse model outperformed this classical baseline by a significant margin. Under the same setup and data provided by Nair et al., our forward consistency based imitator significantly outperforms the model proposed in Nair et al. This directly implies that our method outperforms this classical baseline.
>
> R1: "... place this paper with respect to doing reinforcement learning first ... controlling the exploration."
> => This is a very relevant question. Our overarching aim is to enable robotic systems to perform  complex new tasks from raw visual inputs. Instead of training our robot to perform only one task, we would like to provide the goal task description (i.e., an image depicting the goal) as input to the robot’s policy.
>
> The most general way would be to learn a policy (say using reinforcement learning) that takes as input the images showing the current and goal state and outputs a sequence of actions to reach the goal. There are some major concerns with using RL in the real world: (a) measuring rewards is non-trivial. For e.g., in order to train the robot to configure the rope in shape S or knot, one would need to build classifiers that detect these shapes and use the binary output of classifier as the reward. However, these classifiers will inevitably be imperfect, which in turn would lead to noisy reward. More critically, in order for the system to generalize to novel goals, one would have to train the system with many goals. This implies that large amounts of human supervision would be required to train these classifiers.  (b) RL typically requires ~10-100 million interactions to learn from visual inputs for all but the simplest of tasks, which is simply infeasible in the real world.
>
> In our paper, we propose to learn such a policy using supervised learning. The agent explores its environment and generates interaction data in the form of pair of states and sequence of actions the agent executed. This action sequence provides supervision to learn the policy. One major issue in training such a model through supervised learning is that multiple actions can help the agent transition from current to goal state. We resolve this issue by proposing the forward consistency loss. Our method is sample efficient (~60K interactions for rope manipulation, ~200K for navigation) and does not rely on environmental rewards.
>
> The number of samples required to learn such a policy grows with the number of actions needed to reach goal. To perform complex tasks, we trade off this difficulty by using subgoals in the form of a sequence of images provided by an expert. In other words, we learn a low-level policy (i.e. PSF) which is accurate for predicting actions when current and goal state are not that far apart, and the expert demonstration provides a high-level plan to goto far away goal states.
>
> R1: "pretty relevant paper which attempts the same task ... by Zhu et al."
> => This paper uses RL using multiple goals. One major issue with RL based methods in real world is their bad sample efficiency. Adding multiple goals to the same policy usually hurts the sample efficiency even more, making it generally impractical to train on real robots. For e.g., the above paper trained using RL for 100M samples in simulation for 10-20 steps away goals. However, this is a relevant citation and we will include it in the final revision.

---

> > ### Comment · AnonReviewer1 · 2017-12-29
> > **More clarification questions.**
> >
> > I agree that pure RL can be pretty sample inefficient especially with raw visual input and multiple goals. I also agree that when using pure RL providing reward is generally difficult especially in such visual input cases ((a) in the response above). My question is that how much more/less work is it in general to train the independent goal recognition network via supervised learning?--"In this work, we learn a robust goal recognition network by classifying transitions collected during exploration. We sample states at random, and for every sampled state make positives of its temporal neighbors, and make negatives of the remaining states more distant than a certain margin. We optimize our goal classifier by cross-entropy loss." The burden of providing reward in pure RL is now replaced with the supervision needed to train the goal classifier. Doesn't this step use expert supervision?

---

> > > ### Author Response · Authors · 2017-12-30
> > > **Follow-up response to AnonReviewer1**
> > >
> > > We thank you for following up on the discussion. The independent goal recognition network does not require any extra work concerning data or supervision. The data used to train the goal recognition network is the same as the data used to train the PSF. The only prior we are assuming is that nearby states to the randomly selected states are positive and far away are negative which is not domain specific. This prior provides supervision for obtaining positive and negative data points for training the goal classifier. Note that, no human supervision or any particular form of data is required in this self-supervised process.
> > >
> > > In contrast, for rewards in RL, one would be required to train a separate classifier for "each" goal. Training each classifier will require manually annotated data. Therefore, training multiple classifiers will require substantial human supervision. Furthermore, these classifiers will be noisy, leading to noisy reward which will make the variance of policy gradients even higher. In our self-supervised setup, we do not require any human supervision and noisy learning of goal classifier is okay because we do not use the goal classifier to train the policy.
> > >
> > > Hope this clarifies potential advantages of the proposed self-supervised learning approach in terms of sample efficiency and reward engineering over a pure RL based approach. We will refine the text to make it clearer in the final revision.

---

> > > > ### Comment · AnonReviewer1 · 2018-01-01
> > > > **Thanks! Review upgraded.**
> > > >
> > > > Thanks for the clarifications. I have updated my review grade accordingly.

---

### Author Response · Authors · 2017-12-26
**Response to reviewers**

We thank the reviewers for their insightful and helpful feedback. We are glad that the reviewers found the general idea original and especially relevant (R2); the proposed approach well-founded and the experimental evaluations promising (R3). R3 says "I am very happy to see experimental evaluations on real robots, and even in two different application domains." Both R2 and R3 recommend clear accepts for the paper. R1 asked for comparison to classical methods and a discussion on differences and similarities from pure reinforcement learning based approaches. In direct replies to individual reviewers, we report the performance of the requested baselines, explain the differences from a pure RL based approach, and address remaining questions.

Update:
We thank R1 for taking the time to follow up on our comments with insightful discussion, and upgrading the review score for the paper to accept.

---

### Decision · Program_Chairs · 2018-01-29
**ICLR 2018 Conference Acceptance Decision**

**Decision:**

Accept (Oral)

**Comment:**

The authors have proposed a method for imitating a given control trajectory even if it is sparsely sampled. The method relies on a parametrized skill function and uses a triplet loss for learning a stopping metric and for a dynamics consistency loss. The method is demonstrated with real robots on a navigation task and a knot-tying task. The reviewers agree that it is a novel and interesting alternative to pure RL which should inspire good discussion at the conference.